# Molecular Mapping and Candidate Gene Analysis for GA_3_ Responsive Short Internode in Watermelon (*Citrullus lanatus*)

**DOI:** 10.3390/ijms21010290

**Published:** 2019-12-31

**Authors:** Haileslassie Gebremeskel, Junling Dou, Bingbing Li, Shengjie Zhao, Umer Muhammad, Xuqiang Lu, Nan He, Wenge Liu

**Affiliations:** Zhengzhou Fruit Research Institute, Chinese Academy of Agricultural Sciences, Zhengzhou 450009, China; hailenakirosa@gmail.com (H.G.); junlingdou@163.com (J.D.); newbbing@gmail.com (B.L.); zhaoshengjie@caas.cn (S.Z.); mjawadumer@gmail.com (U.M.); luxuqiang@caas.cn (X.L.); henan@caas.cn (N.H.)

**Keywords:** candidate gene, BSA-Seq, GA 3β-hydroxylase, fine mapping, cytological analysis

## Abstract

Plants with shorter internodes are suitable for high-density planting, lodging resistance and the preservation of land resources by improving yield per unit area. In this study, we identified a locus controlling the short internode trait in watermelon using Zhengzhouzigua (long internode) and Duan125 (short internode) as mapping parents. Genetic analysis indicated that F_1_ plants were consistent with long internode plants, which indicates that the long internode was dominant over the short internode. The observed F_2_ and BC_1_ individuals fitted the expected phenotypic segregation ratios of 3:1 and 1:1, respectively. The locus was mapped on chromosome 9 using a bulked segregant analysis approach. The region was narrowed down to 8.525 kb having only one putative gene, *Cla015407*, flanking by CAPS90 and CAPS91 markers, which encodes gibberellin 3β-hydroxylase (GA 3β-hydroxylase). The sequence alignment of the candidate gene between both parents revealed a 13 bp deletion in the short internode parent, which resulted in a truncated protein. Before GA_3_ application, significantly lower GA_3_ content and shorter cell length were obtained in the short internode plants. However, the highest GA_3_ content and significant increase in cell length were observed in the short internode plants after exogenous GA_3_ application. In the short internode plants, the expression level of the *Cla015407* was threefold lower than the long internode plants in the stem tissue. In general, our results suggested that *Cla015407* might be the candidate gene responsible for the short internode phenotype in watermelon and the phenotype is responsive to exogenous GA_3_ application.

## 1. Introduction

Watermelon (*Citrullus lanatus* (Thumb) Matsum and Nakai) belongs to the genus *Citrullus*, is one of the most important horticultural crop in the world [1,2,3,4]. The genus *Citrullus* includes about 118 genera and 825 species for diploid species (2*n* = 22), which are grown in Africa, Asia and in the Mediterranean region [4,5]. Watermelon is an important member of Cucurbitaceae family that is commercially cultivated worldwide and currently China is by far the world’s largest producer of watermelon, followed by Turkey and Iran, 79.2, 3.9 and 2.8 million tons, respectively (https://www.worldatlas.com, accessed on 12 April 2019). Watermelon has a small genome: 4.25 × 10^8^ base pairs for the diploid chromosomes [6,7]. The watermelon genome has been sequenced and 23,440 predicted protein-coding genes were identified [8,9]. From the genetic studies, more than 100 genes have been identified [7] that are involved in seed and seedling, vine, flower and fruit as well as resistance to diseases, insects, and stress traits.

Dwarf genes played a vital role during the ‘Green Revolution’, which was one of breakthroughs in yield improvement via the introgressive hybridization of dwarf and semi-dwarf traits into cereal crop cultivars [10,11]. Dwarf genes are associated with optimizing yield [12], dense field cultivation [13], lodging resistance [14], decrease damages due to wind and rain [15], early maturity [16], increase tillering capacity [17,18], and used in modern plant breeding programs. In watermelon breeding programs, dwarf genes with short internode have significant advantages in reducing the labor required for cultivation, suitable for high density planting and save land resources by improving the yield per unit area [19].

In crop plants, dwarfism mainly occurred due to gene mutation related to hormonal biosynthesis and signaling pathways [20,21]. The mutation of genes in GA biosynthesis pathway caused GA deficiency, which resulted in a severe dwarf phenotype in plants [22,23]. The predominant semidwarf1 (*sd1*) gene in rice cultivars caused dwarf phenotype due to the deficiency of GA20-oxidase activity [24]. The dwarf genes in wheat *Rht-B1b* and *Rht-D1b* were caused due to the mutation in the GA signaling pathway, *Rht18* due to increased *GA2-oxidaseA9*expression [25,26], and the shortened basal internodes (*SBI*) in rice was due to decreased gibberellin 2-oxidase activity [27]. Likewise, the *btwd1* [28] in barley, *Dw1* [29] in sorghum and a recessive *d2* [30] in pearl millet were reported as mutants in the GA biosynthesis associated with dwarf phenotype.

In cucurbit crops, several recessive genes have been reported to confer the short internode length. The bushy growth habit in pumpkin was characterized by short internode [18,31] and showed monogenic inheritance, in which the bushy genotype is dominant (*Bu*) to the vine genotype (*bu*) [32]. In cucumber, *cp* gene loci was associated with short internodes and shorter growing period [33,34]. Four recessive dwarfing genes, *si-1*, *si-2*, *si-3*, and *mdw1*, were reported to determine the short internode [35,36] in melon. In *Cucurbita pepo* and squash (*Cucurbita* spp.), a dominant *Bu* locus was associated to short internode phenotype [37]. In watermelon, four single recessive gene loci, *dw-1*, *dw-2*, *dw-3* and *dw-4* were associated with short internode and dwarf vine with bushy growth habit [5]. Furthermore, a candidate gene (*Cla010726*) encodes gibberellin 20-oxidase-like protein was responsible for dwarfism in watermelon [19].

In the GA biosynthetic pathway of higher plants, GA 3β-hydroxylase catalyzes the conversion of inactive gibberellin precursors (GA_20_, GA_5_ and GA_9_) to bioactive gibberellins (GA_1_, GA_3_ and GA_4_), respectively [38,39]. Bioactive GAs are synthesized by 3-beta-hydroxylation and catalyzed by GA 3β-hydroxylase enzyme in the presence of 2-oxoglutarate binding region, which is essential for its activity [40]. Several GA 3β-hydroxylase encoding genes have been cloned from different plant species [41]; including barley [42], rice [43] and Arabidopsis [44]. Moreover, GA 3β-hydroxylase from pumpkin endosperm catalyzes both 2-beta and 3β-hydroxylation [45] and in maize *dwarf-1*, catalyzes three hydroxylation steps in GA biosynthesis pathway [46] indicating that GA 3β-hydroxylase has a multifunctional activity.

The development of high-throughput sequencing technologies and availability of the watermelon reference genome [8], genomic variation maps [47], bulked-segregant analysis [48], and next generation sequencing (NGS) technologies [49] significantly accelerated the identification of candidate genes controlling important agronomic traits. Bulked segregant analysis (BSA) is an important method for rapidly identifying gene or genomic regions on chromosomes that are linked to a causative mutation in a group of phenotypically mutant plants [50,51]. This method works with selected and pooled individuals that has been extensively used in gene mapping with bi-parental populations, mapping by sequencing with major gene mutants and pooled genome wide association study using extreme variants [52]. Although many research works on dwarfing traits in watermelon had been done before, the underlying molecular mechanisms and identification of genes responsible for short internode was not yet identified. Therefore, this experiment was designed to characterize the inheritance of the short internode in Duan125, identified the genomic region through bulked segregant analysis sequencing (BSA-Seq) approach, and a candidate gene controlling the short internode phenotype in watermelon.

## 2. Results

### 2.1. Agronomic Characteristics and Inheritance of the Short Internode

The Zhengzhouzigua (long internode) plants had longer internode length (cm), more number of internodes, longer vine length (m) (Figure 1A,E, Appendix A) compared with Duan125 (short internode) plants (Figure 1B,H, Appendix A).

The highest internode length (5.50 cm), number of internodes (26.50) and vine length (2.04 m) were obtained from the long internode plants. In contrast, the lowest values were obtained from the short internode plants (Figure 2A–C). In the short internode plants, internode length, number of internodes and vine length were significantly reduced by 42.73%, 49.70%, and 57.84%, respectively, compared to long internode plants.

In this study, we successfully developed F_2_ and BC_1_ populations from Zhengzhouzigua (long internode) and Duan125 (short internode) as mapping parents in 2016, 2017 and 2018 (Henan) and in 2017 (Hainan). Phenotypes of F_1_ plants exhibited long internode confirming that the long internode is dominant over the short internode. The observed segregation in the F_2_ individuals showed that 278, 325 and 367 plants were segregated as long internode, while 89, 105 and 119 plants were with short internode in 2016 (winter), 2017 and 2018 (spring) seasons in Henan, which fitted the expected segregation ratio of 3:1 (χ^2^ = 0.110, *p* = 0.740; χ^2^ = 0.078, *p* = 0.781; χ^2^ = 0.069, *p* = 0.79), respectively. Moreover, 431 plants were segregated as long internode, while 147 plants were with short internode in 2017 (winter) in Hainan, which fitted the expected segregation ratio of 3:1 (χ^2^ = 0.058, *p* = 0.81). None of the BC_1_P_1_ individuals obtained from a backcross between F_1_ and long internode plants were short internodes. However, for the backcross of F_1_ with short internode (BC_1_P_2_), 38 plants had long internodes while 42 plants had short internodes, which fitted a segregation ratio of 1:1 (χ^2^ = 0.200, *p*= 0.6547) (Appendix A). These results confirm that the short internode is controlled by a single gene with the short internode phenotype being recessive in watermelon.

### 2.2. BSA-Seq Analysis Identified a Candidate Gene Located on Chromosome 9

The whole genome resequencing (WGR) through BSA-seq approach was used for identifying the genomic region contributing to the short internode phenotype. A total of 30.1 Gb raw data having approximately 30 × depth and more than 99% coverage for each pool were generated. After filtering out adaptor and low quality reads, the clean reads were aligned to ‘97103′ watermelon reference genome (http://cucurbitgenomics.org/organism/v1, accessed on 5 June 2019), to identify high quality SNPs between the long internode and short internode pools. From the L-pool and S-pool, 102,832,531 and 108,242,182 short reads were generated through the high-throughput sequencing, respectively. Furthermore, after clean reads were aligned to watermelon reference genome, 304,565 SNPs were obtained between the two mixed pools. Finally, using the ΔSNP index strategy, we identified a candidate region on chromosome 9 (Figure 3A). Therefore, these results indicated that there was a locus, designated *dw*, responsible for the short internode trait located between 9.433 kb to 34.42 Mb on chromosome 9.

### 2.3. Fine Mapping, Annotation and Candidate Gene Sequencing

Cleaved amplified polymorphic sequence (CAPS) markers were developed based on the SNPs between the two parental lines using the ‘97103’ watermelon reference genome (http://cucurbitgenomics.org/organism/v1, accessed on 19 January 2019) (Appendix A) for polymorphic screening. The mapping interval was narrowed down to a physical distance of 8.525 kb (1,850,884–1,859,409 kb) between CAPS90 and CAPS91 markers with 12 recombinant individuals (Figure 3B, Appendix A). In this mapping interval, only one putative (*Cla015407*) gene having two exons and one intron was annotated using the online ‘97103′ watermelon reference genome (Figure 3C). We designed gene specific primers (Appendix A) to amplify both the genomic and entire coding sequences (CDS) of both the parental lines. The sequences alignment indicated that a 13 base pair deletion (502–514 bp) was found in the second exon of the short internode parent (Figure 3D). The 13 bp deletion resulted in a frameshift mutation, which led to truncated protein. The gene annotation in ‘97103’ watermelon reference genome revealed that *Cla015407* encodes GA 3β-hydroxylase. GA 3β-hydroxylase is an important enzyme in the downstream of GA biosynthesis pathway, which catalyze the inactive precursors of GA_9_, GA_20,_ and GA_5_ into bioactive GA_4_, GA_1,_ and GA_3_, respectively.

### 2.4. Homology, Phylogenic Tree and Conseved Domain Analysis

The BLAST result in TAIR (https://www.arabidopsis.org, accessed on 23 May 2019) indicated that *Cla015407* was highly homologous to *AT1G15550* in *Arabidopsis thaliana*, which encodes GA 3β-hydroxylase enzyme. This enzyme is involved in the later steps of the gibberellic acid biosynthetic pathway activated by AGAMOUS in a *cal-1* and *ap1-1* background [53,54]. To understand the relationship between *Cla015407* protein sequences and other homologous, we BLAST the protein sequence of *Cla015407* in both NCBI (http://www.ncbi.nlm.nih.gov/, accessed on 24 May 2019) and Uniprot database (https://www.uniprot.org/, accessed on 23 May 2019). The phylogenetic tree was generated using the neighbor-joining method as implemented in MEGA7 software through bootstrap method with 1000 replications [55]. The result indicated that *Cla015407* gene has a close relationship and shares a common ancestor with *XP004135830* from *Cucumis sativus* and *XP008461056* from *Cucumis melo* (Figure 4A). This indicated that the *Cla015407* was evolutionarily conserved within the Cucurbitaceae family. Furthermore, we generate the protein domain structure for *Cla015407* using the online Pfam database (http://pfam.xfam.org/, accessed on 23 December 2019). The sequence alignment using UniProt and SMART indicated that *Cla015407* was shared 57.10% sequence identity with *AT1G15550* in *Arabidopsis thaliana*, which containing two domains: DIOX_N and 2OG-Fell_Oxy [44]. The deletion in the CDS region of short internode watermelon caused a premature stop codon, producing a truncated protein with only 173 amino acid residues, losing the 2OG-Fell_Oxy domain (Figure 4B).

### 2.5. Gene Expression Analysis

To analyze the expression levels of *Cla015407* in both long and short internode parents, gene specific primers (Appendix A) were designed. The expression pattern of *Cla015407* indicated that it was expressed in all tissue parts, most prominently in the stem (internode) part. In the long internode plants, the expression level of *Cla015407* was much higher (2.99) in stem followed by root (0.60) and leaf (0.54) tissue parts. Whereas in the short internode parent, transcript level in the stem tissue was 3.56 fold lower than in the long internode parent (Figure 5). These results revealed that the expression level of *Cla015407* in short internode parent was significantly reduced compared with the long internode parent in the stem, and this result further confirmed that *Cla015407* might be the candidate gene controlling the short internode phenotype in watermelon.

### 2.6. Determination of GA_3_ Hormone in the Short Internode

Phenotypically, internode and vine length were increased in the short internode plants after GA_3_ application (Figure 1E, Appendix A). The internode length (5.69 cm) and vine length (1.88 m) was increased in the short internode plants (Figure 1D,G and Figure 2E,F); however, there was no significant difference compared with long internode plants (Figure 1C,F and Figure 2E). These results showed that the short internode phenotype was restored after exogenous GA_3_ application and was a GA_3_ responsive short internode phenotype. Moreover, to determine the amount of GA_3_, samples were collected from top, middle and basal internode positions of both inbred lines. Before GA_3_ application, significantly higher GA_3_ content (11.16 ng.g^−1^ FW) was obtained at middle internode position of the long internode plants (Figure 6A). In contrast, the lowest GA_3_ content (6.76 ng.g^−1^ FW) was obtained at the top internode position of the short internode plants. After exogenous GA_3_ application, the significantly higher GA_3_ content (20.88 ng.g^−1^ FW) was found at the top internode position of the short internode plants, while the lowest (15.69 ng.g^−1^ FW) value was obtained at the same internode position in the long internode plants (Figure 6B). Overall, GA_3_ content was significantly reduced before GA_3_ application; however, it was increased after exogenous GA_3_ application in the short internode plants, suggesting that the loss of function of *Cla015407* caused an impaired GA 3β-hydroxylase enzyme activity, and thus led to short internode phenotype.

### 2.7. Microscopic Observation and Verification of the Short Internode

To observe the difference in length and size of cells between short and long internode plants, samples were collected from three different internode positions (top, middle and basal). Before GA_3_ application, the length and thickness of cells in the short internode plants were significantly reduced (Figure 7G–I) compared with long internode plants (Figure 7A–C) in all internode positions. In the short internode plants, the cell length in the basal internode position was reduced; whereas, the cell size was increased (Figure 7I) compared with long internode plants (Figure 7C). After GA_3_ application, the length and size of cells in the short internode plants were increased (Figure 7J,K), and had no significant difference with long internode plants in all internode positions (Figure 7D,E) except in the basal internodes (Figure 7F,L). Therefore, these results further suggested that loss of function of *Cla015407* led to shorter cells and internode length in watermelon.

In order to verify whether the 13 bp deletion in *Cla015407* was the causal variation for the short internode phenotype, we develop an InDel marker based on the 13 bp sequence of *Cla015407* gene (Appendix A). Then to validate this marker, 135 F_2_ individuals including the 12 recombinants were selected to confirm the genotype. The results indicated that 33 individuals were homozygous dominant and 67 individuals were heterozygous, while 35 individuals were homozygous recessive. This result confirmed that the phenotype was in harmony with the genotype (Figure 8). Overall, these results indicated that *Cla015407* could be the candidate gene that controls the short internode phenotype in watermelon.

## 3. Discussion

The development of dwarf varieties having short internode with improved mechanical stability of stems prevents lodging, leading to significant increase in crop productivity. Miniature dwarf type in cucurbits may provide an alternative to standard vining types for plastic tunnel production due to their amenability to low maintenance upright cultivation [36]. The dwarf vine watermelon with short internode has a significant advantage in enhancing yield per unit area, lodging resistance (for upright vine growth habit), reduces the labor for cultivation, is suitable for high density planting, and saves land resources. Internode length and vine length are significantly reduced in ‘*dsh*’ dwarf mutant than the ordinary watermelon [5]. In soybean, both internode length and plant height were reduced in *dw* mutant compared to wild type [56]. In pear millet, the shorter internode was reduced, which resulted in total reduction in plant height [57]. In this study, the short internode plants showed significantly reduced internode length (cm), number of internodes, and vine length compared to the long internode (Figure 1 and Figure 2).

Forward genetics can reveal new genes and advanced view of gene function essential for isolating candidate genes of important traits [56]. It is also useful to understand how genes and gene networks contribute to build an organism [58,59,60]. Several mapping strategies based on next generation sequencing (NGS), such as direct resequencing, have been developed for the rapid detection of causal mutations controlling target traits [36]. Nowadays, the bulked-segregant analysis (BSA-Seq) approach is a more popular strategy for cloning mutant genes in various crops [52]. It helps in identifying a mapping interval and candidate single nucleotide polymorphisms (SNPs) from whole genome sequencing of pooled F_2_ individuals [59]. In watermelon, four single recessive gene loci, *dw-1*, *dw-2*, *dw-3* and *dw-4* were reported to confer the short internode and dwarf vine with bushy growth habit [5,61]. The *cp*, *cp-2*, and *scp* genes had been identified for dwarfism related plant architecture in cucumber [62,63], whereas in tropical pumpkin and squash the dwarf vine was regulated by *Bu* gene [37]. In this study, we identified a locus (*dw*) responsible for short internode on chromosome 9 through BSA-Seq approach (Figure 3A) and the candidate gene (*Cla015407)* corresponding to short internode was delimited to 1850884–1859409 kb region between CAPS90 and CAPS91 markers (Figure 3B). The sequence alignment between the two parental lines indicated that a 13 bp deletion was identified in the CDS region of *Cla015407* in short internode parents causing a frameshift mutation (Figure 3C). The expression of *Cla015407* gene in the short internode parent was significantly reduced as compared to long internode parents; however, there was no significant difference in leaf and root parts.

Dwarfing genes associated with gibberellin acid (GA) were categorized as gibberellin insensitive (GAI) and gibberellin responsive (GAR) based on their effect on plant height or internode length. The reduction in plant height by dwarf genes varies with different genetic backgrounds [64,65,66]. The double dwarf genes produced more shorter internode length than single dwarf genes [66,67]. The GAI dwarfing genes have been widely used to reduce plant height and increase grain yield in crop breeding programs. GAI dwarfing genes *Rht-B1b (Rht1)* and *Rht-D1b (Rht2)* were reported in wheat that reduced internode length and the overall plant height [68]. In contrast, in GAR genes, shorter internode plants respond positively to exogenous GA hormone and were grown similarly as a normal plants [69]. Mutants deficient in GA biosynthesis can be rescued by exogenous application of bioactive GAs; however, it is not possible if the mutation is in the GA signaling pathway [70,71]. A number of GAR dwarf mutants has been isolated from various plant species, including watermelon, maize, pea, tomato, Arabidopsis, rice and wheat [5,68,72,73,74]. In this study, the internode length and vine length increased in the short internode plants after exogenous GA_3_ application.

Different plant hormones interact each other hormones [75,76] and coordinately control the cell elongation, cell proliferation and cell differentiation [77]. Brassinosteroids (BR) related mutants display pleiotropic dwarf phenotypes caused by defects in cell elongation and differentiation processes, which determine the cell length [66,78]. In rice [79] and cotton [80], the mutation of GA biosynthesis related genes inhibits cell elongation, indicating that GA has an important role in internode cell elongation. In this study, to elucidate the length and size of cells, we conduct cytological analysis of both long and short internode plants at different internode positions. Before GA_3_ application, the length and size cells were significantly reduced in short internode plants. However, the length and size of cells in the short internode plants were similar with long internode plants after exogenous GA_3_ application at top and middle internode positions. In the top internode positions, elongated and un-thickened cells were observed; whereas, modestly elongated and thickened cells were observed in the middle internodes. However, basal internodes had well developed tissues, shorter length, and more thick cells than middle and top internode positions in both long and short internode plants. GA_3_ is important in determining plant height by regulating internode cell elongation. In rice, a positive regulator of both GA biosynthesis and GA signaling *AtERF11* gene is associated with internode cell elongation and promotes by increasing bioactive GA_3_ accumulation [81]. Exogenous GA_3_ application enhanced internode cell elongation in pea that the dwarf cultivar responds positively to exogenous GA_3_ [82]. Furthermore, other studies have shown that GA_3_ can promote cell elongation in internode tissue and act as regulators of stem elongation [83,84]. GA_3_ induced cell elongation should have great significance to rice plants, which in turn develop a low-sensitivity pathway in response to high hormone levels [85].

Our results revealed that the loss of function of *Cla015407* impaired GA 3β-hydroxylase enzyme activity, which led to a reduced GA_3_ content, shorter cells, and shorter internode length in the short internode plants. In general, we identified a candidate gene (*Cla015407)* responsible for short internode length encoding GA 3β-hydroxylase enzyme that catalyzes the conversion of inactive GA_20_, GA_9_ and GA_5_ to bioactive GA_1_, GA_4_ and GA_3_, respectively. This research finding will be helpful for the marker-assisted selection and molecular regulatory mechanisms of short internode length in watermelon.

## 4. Materials and Methods

### 4.1. Plant Materials and Mapping Population

In this study, two inbred lines Zhengzhouzigua (LI: long internode) and Duan125 (SI: short internode) were used as mapping parents. The short internode plants had short internodes, fewer numbers of internodes, shorter vine length, bushy growth habit and shorter internode cell length than the long internode plants. The seeds of both inbred lines were obtained from Zhengzhou Fruit Research Institute, National Watermelon and Melon Germplasm Resource Library, Henan, China. F_1_ plants were obtained after crossing the two parental lines and then F_2_ population was developed through selfing F_1_ plants. Backcross population was also obtained by hybridizing F_1_ with each of the parental lines to develop BC_1_P_1_ (F_1_ X Zhengzhouzigua) and BC_1_P_2_ (F_1_ X Duan125). For genetic mapping and segregation analysis, F_2_ population were grown at two locations and three different years: Xinxiang experimental site in 2016 (Henan, winter), 2017 and 2018 (Henan, spring) and Sanya experimental site in 2017 (Hainan: spring) seasons. The two parental lines, F_1_, 70 BC_1_P_1_ and 80 BC_1_P_2_ individuals, were grown in spring 2018 at Xinxiang experimental site Henan, China (Appendix A). The short internode trait was evaluated for phenotypic analysis and data on internode length, number of internodes and total vine length were measured and analyzed using analysis of variance (ANOVA) using general linear model (GLM) statistical analysis software programs.

### 4.2. DNA Extraction and Bulked Segregant Analysis

Genomic DNA from the young leaves of 30 long and 30 short internodes of F_2_ individuals and the two inbred lines were isolated using plant genomic DNA kit (TIANGEN, Beijing, China). Whereas, the genomic DNA from 430 F_2_ population was extracted using CTAB (Hexadecyl Trimethyl Ammonium Bromide) method as previously reported [86] with minor modifications from the original method. Equal amount of DNA (5-µg) from 30 long and 30 short internodes were pooled together to construct long internode pool (L-pool) and short internode pool (S-pool), respectively. The two DNA pools were used for BSA-sequencing and a pair end sequencing libraries having read length of 100 bp with insert size approximately 500 bp were prepared for sequencing using an Illumina HiSeq2000 machine. The sequence reads from L-pool and S-pool were aligned to the ‘97103’ watermelon reference genome (http://cucurbitgenomics.org/organism/v1, accessed on 21 January 2019) using BWA software (Wellcome Trust Sanger Institute, Wellcome Genome Campus, Cambridge, CB10 1SA, UK.) [87]. The aligned files were converted to SAM/BAM files using SAM tools [88] and then applied to SNP calling filter [89,90]. The SNP index was assigned as zero, i.e., all the short reads were from the ‘97103’ watermelon reference genome. The average SNP indices of the SNPs located in a given genomic region were calculated using sliding window analysis with a 1 Mb window size and a 10 kb increment. The SNP index graphs for each L-pool, S-pool, and ΔSNP index were plotted to identify the genomic region containing the short internode locus. The ΔSNP index was calculated by subtracting the S-pool index from the L-pool index [91,92]. Computer simulation was performed to generate confidence intervals of the SNP index value under the null hypothesis of no locus. The ΔSNP index was derived from the calculated SNP index for each pool and this process was repeated 10,000 times for each read depth, and confidence intervals were produced.

### 4.3. Fine Mapping through CAPS Markers

To narrow down the genomic region and identify candidate gene responsible for short internode, cleaved amplified polymorphic sequences markers (CAPS) were developed based on the SNPs generated from BSA-Seq method (Appendix A). These markers were used to screen the 430 F_2_ population developed by selfing from long and short internode plants. The PCR reaction mixture for CAPS amplification was carried out in a total volume of 10 μL containing 1μL (10 ng) of genomic DNA, 0.5 μL for each forward and reverse primers (10 pmol/μL), 1 μL 10 × PCR buffer, 0.2 μL dNTPs (10 mM), and 0.1 μLTaq polymerase (5 units/μL) or 5 μL 2 × Power Taq PCR Master Mix. PCR was performed by pre-heating samples for 5 min at 95 °C followed by 30 cycles of 30 s at 95 °C, 30 s at 57 °C, and 50 s at 72 °C, finishing with post-heating for 10 min at 72 °C. Then, the PCR products were digested by restriction endonuclease enzyme to verify polymorphisms of the CAPS markers. The reaction mixture for enzyme digestion was contained 10 μL PCR product and 5 μL total enzyme reaction solution having 3.3 μL ddH_2_O, 1.5 μL 10 × buffer and 0.2 μL restriction enzyme which was incubated at 37 °C, 50 or 65 °C for 5–16 h depending on the instructions for the restriction enzymes. Finally, the enzyme-digested products were separated in 1% agarose gel electrophoresis.

### 4.4. Annotation, Cloning and Sequencing Analysis of Candidate Genes

After narrowing down of the mapping interval through CAPS markers, the candidate genes in target region were predicted using the online ‘97103’ watermelon reference genome database (http://cucurbitgenomics.org/organism/v1, accessed on 25 May 2019) and their annotations were obtained with the BLASTP in NCBI (https://blast.ncbi.nlm.nih.gov/Blast.cgi, accessed on 26 May 2019). Gene specific primers were designed and the full length of genomic DNA and entire coding sequence (CDS) were amplified from both long and short internode inbred lines. The PCR amplification was carried out according to the user manual with 2 × Phanta Max Master Mix (Vazyme Biotech, Nanjing, China). The amplicons were separated on 1% agarose gel, cloned and purified with a TIANgel midi DNA Purification Kit (TIANGEN, Beijing, China) according to the manufacturer’s instructions. The purified products were then subjected to a “^+^ A base reaction using a kit (ZOMANBIO, Beijing, China), introduced in to the vector pMD19-T with a pMD™18-T Vector Cloning Kit (TaKaRa, Tokyo, Japan), and transformed into DH5α chemically competent cell (*E. coli* strain DH5α) (Weidibio, Shanghai, China) according to the manufacturer’s protocol. All the fragments were sequenced by Sangon Biotech Co., Ltd. (Shanghai, China).

### 4.5. Conserved Domain, Phylogenetic Analysis and Verification of the Short Internode Using an InDel Marker

The protein sequence alignment was performed using Uniprot (https://www.uniprot.org/, accessed on 27 May 2019) and SMART (https://blast.ncbi.nlm.nih.gov/smartblast/, accessed on 27 May 2019, databases to illustrate the domain structure. Phylogenetic analysis was performed using MEGA 7 software (The Pennsylvania State University, University Park, PA, USA) with a bootstrap method and 1000 replications [93]. Moreover, to validate the causal mutation, an InDel marker was developed conferring to the 13 bp sequence of *Cla015407* and 135F_2_ progenies including the recombinant individuals were selected to check the polymorphism of the marker.

### 4.6. Expression Analysis of Candidate Gene

To analyze the gene expression; samples were collected from root, stem and leaf parts of both long and short internode parents. Total RNA was isolated using plant RNA purification kit (TIANGEN, Beijing, China) following the manufacturer’s protocol and treated with RNase free DNase) to remove residual genomic DNA. Complementary DNA (cDNA) was synthesized with reverse transcriptase M-MLV (RNase) following the manufacturer’s instructions (TaKaKa, Tokyo, Japan). Primers for candidate gene and reference gene used in quantitative reverse transcription polymerase chain reaction (qRT-PCR) were designed based on ‘97103’ watermelon reference genome (http://cucurbitgenomics.org/organism/v1, accessed on 7 June 2019). Expression levels of the target gene were evaluated by qRT-PCR using a LightCycler480 RT-PCR system (Roche, Basel, Switzerland). All reactions were performed using the SYBR Green real-time PCR mix according to the manufacturer’s instructions. Each 20 μL RT-PCR reaction mixture containing 1 μL cDNA, 1 μL forward primer (10 μM), 1 μL reverse primer (10 μM), 10 μL 2x SYBR Green real-time PCR mix, and 7 μL nuclease-free water was pre-heated at 95 °C for 5 min, followed by 45 cycles of 95 °C for 30 sec and 72 °C for 30 sec. High-resolution melting was performed at 95 °C for 1 min, at 40 °C for 1 min, at 65 °C for 1 min, and continuous at 95 °C. All experiments were performed in three biological replicates. The raw data obtained from qRT-PCR was analyzed using LCS480 software 1.5.0.39 (Roche, Basel, Switzerland) and the relative expression was determined using the 2^−ΔΔCT^ method [94].

### 4.7. Application of Exogenous GA_3_ Hormone

A stock solution of GA_3_ hormone having 200 mg/L was sprayed two times a week after seven days from the date of transplanting and the control was treated with equivalent amount of distilled water. After twenty days of the second GA_3_ application, samples were collected from (3th), (11th) and (19th), as well as (3rd), (7th), and (10th) internode positions starting from the top part for both long and short internode plants, respectively. The amount of GA_3_ hormone was determined using enzyme-linked immunosorbent assays (ELISA) method (College of Agriculture and Biotechnology, China Agricultural University, Beijing, China) with three biological and three technical replicates for each set of treatments.

### 4.8. Cytological Analysis of the Short Internodes

To understand the difference in length and size of cells, internode samples were collected from top, middle and basal internode positions of both long and short internode plants. The samples were immediately fixed in FAA (3.7% formaldehyde, 5% glacial acetic acid and 50% ethanol) under vacuum for 24 h. The samples were subsequently dehydrated in a graded ethanol series (70%, 85%, 95% and 100%), infiltrated with xylene and embedded in paraffin with an Epon812 (Henan Chinese Science and Technology, Henan, China). Ultrathin longitudinal sections were sliced using an ultramicrotome, mounted on slides, and stained with 0.05% Toluidine Blue O. The internodes cell elongation was examined with an OLYMPUS BX51 light microscope (Hitachi, Ibaraki, Japan) for 5 min of staining at 10× magnification.

## 5. Conclusions

The *dw* locus was located on chromosome 9 using bulk segregant analysis (BSA-Seq) and the mapping region was narrowed down to the 8.525 kb region. The identified candidate *Cla015407* gene has related function with short internode length encoding GA 3β-hydroxylase. The enzyme is involved in the downstream GA biosynthetic pathway, particularly in the conversion of inactive GA_20_, GA_9_, and GA_5_ to bioactive GA_1_, GA_4_, and GA_3_, respectively. This study will provide a useful reference for understanding the molecular mechanism of short internode, GA biosynthesis pathway, cloning of candidate genes, and the development of short internode watermelon cultivars using marker-assisted breeding.

## Figures and Tables

**Figure 1 ijms-21-00290-f001:**
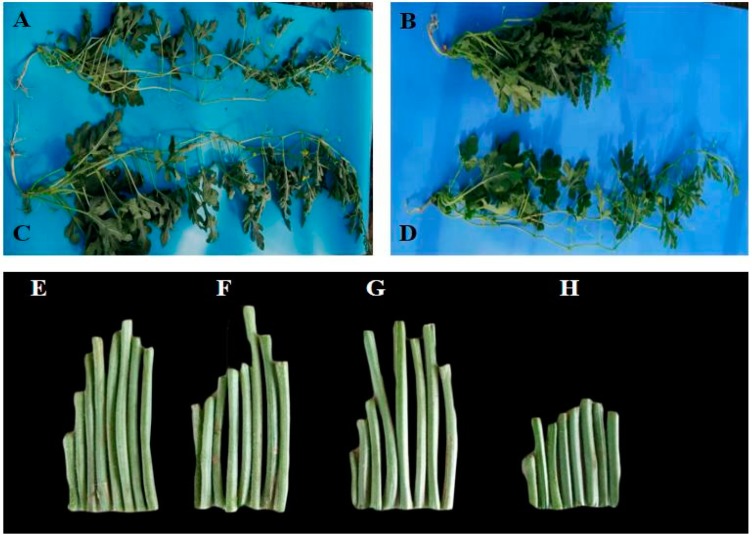
Phenotypic characteristics of Zhengzhouzigua (long internode) and Duan125 (short internode) parents. (**A**) Long internode, and (**B**) Short internode plants before GA_3_ application. (**C**) Long internode, and (**D**) Short internode plants after GA_3_ application. (**E**) Internodes of the long internode plants before GA_3_ application, (**F**) Internodes of the long internode plants after GA_3_ application. (**G**) Internodes of the short internode plants after GA_3_ application, and (**H**) Internodes of the short internode plants before GA_3_ application.

**Figure 2 ijms-21-00290-f002:**
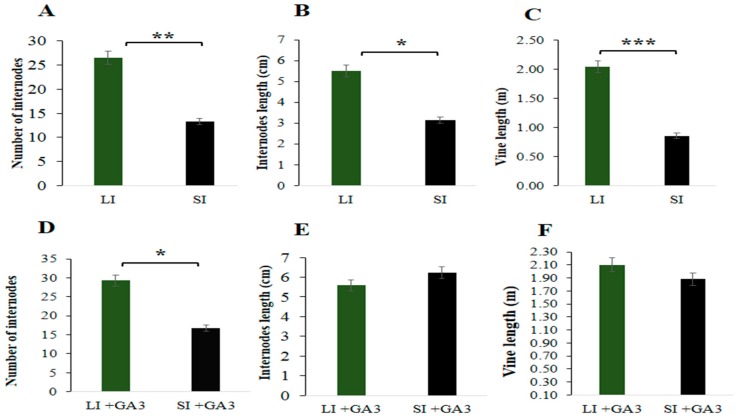
Agronomic traits of long and short internode plants. (**A**) Number of internodes, (**B**) Internodes length (cm), and (**C**) Vine length (m) before GA_3_ application. (**D**) Number of internodes, (**E**) Internodes length (cm) and (**F**) Vine length (m) after GA_3_ application. Data were averages of three biological replications taken from three plants. LI = Long internodes, and SI= Short internodes before GA3 application. LI + GA_3_ = Long internodes, and SI + GA3 = Short internodes after GA_3_ application. Error bars indicates standard deviations from three repeats (*n* = 3). Values are means + SD (*n* = 3). * significant at *p* < 0.05; ** significant at *p* < 0.01 and *** significant at *p* < 0.001 probability levels.

**Figure 3 ijms-21-00290-f003:**
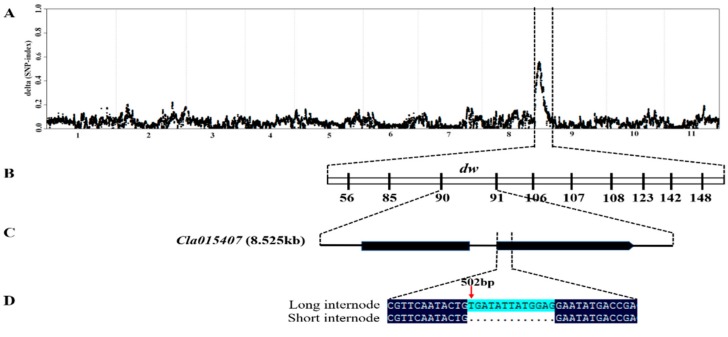
Fine mapping and isolation of the *dw* gene. (**A**) ΔSNP index graph of BSA-seq analysis, (**B**) The *dw* gene was narrow down to an interval 8.525 kb using 430 F_2_ individuals flanking by CAPS90 and CAPS91 markers, (**C**) The gene structure of *Cla015407*. (**D**) Sequence alignment between long and short internode parents showed a 13 bp deletion in the short internode parent.

**Figure 4 ijms-21-00290-f004:**
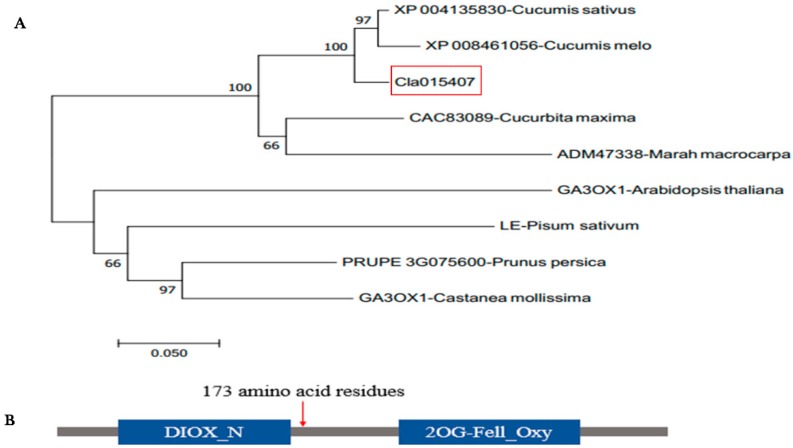
The phylogenetic analysis and conserved domains of the candidate gene. (**A**) Phylogenetic tree for the *Cla015407*. The tree was constructed using MEGA 7 with Bootstrap values calculated from 1000 replicates. The *Cla015407* is circled in red. (**B**) The conserved domain of *Cla015407* gene, which was analyzed by online Pfam database.

**Figure 5 ijms-21-00290-f005:**
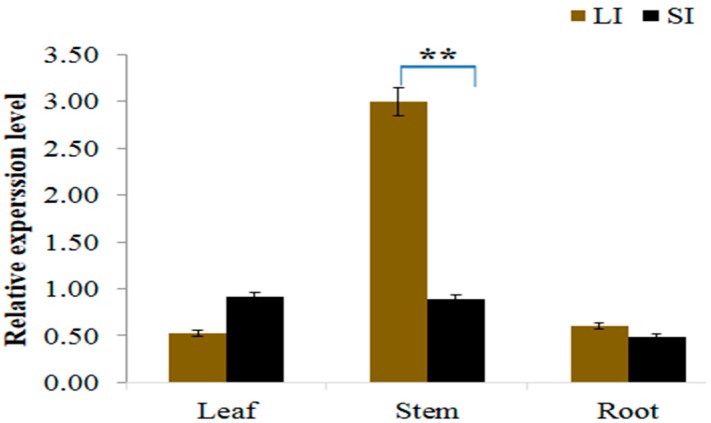
Relative expression level of *Cla015407* in different tissues of both long and short internode parents, Error bars indicates standard deviations from three repeats (*n* = 3). Values are means + SD (*n* = 3). LI = Long internodes, SI = Short internodes. ** Significant at *p* < 0.05 probability level.

**Figure 6 ijms-21-00290-f006:**
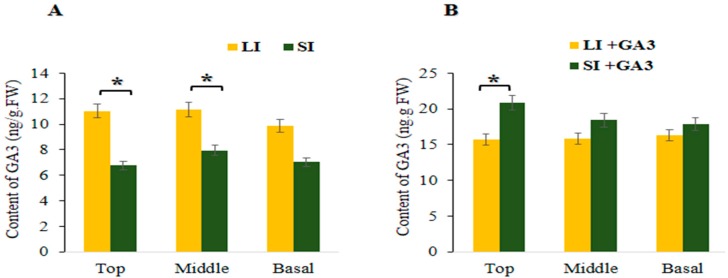
GA_3_ content in top, middle and basal internodes of long and short internode plants. (**A**) GA_3_ content before GA_3_ application, (**B**) GA_3_ content after GA_3_ application. LI = Long internode, and SI = Short internode before GA_3_ application; LI + GA_3_ = Long internode, and SI + GA_3_ = Short internode after GA_3_ application. Error bars indicates standard deviations from three repeats (*n* = 3). Values are means + SD (*n* = 3). * significant at *p* < 0.05 probability level.

**Figure 7 ijms-21-00290-f007:**
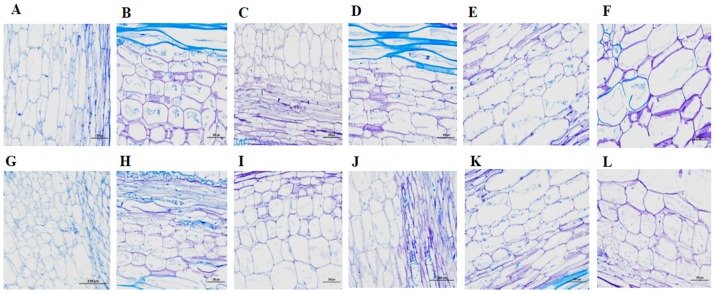
Cytological observation of internode length and size of cells between long and short internode plants. (**A**–**C**) The internode cell length of long internode plants, and (**G**–**I**) The internode cell length of short internode plants from top, middle and basal internode positions, respectively before GA_3_ application. (**D**–**F**) The internode cell length of long internode plants and (**J**–**L**) The internode cell length of short internode plants from top, middle, and basal internode positions, respectively after GA_3_ application. Bar = 100 μm.

**Figure 8 ijms-21-00290-f008:**
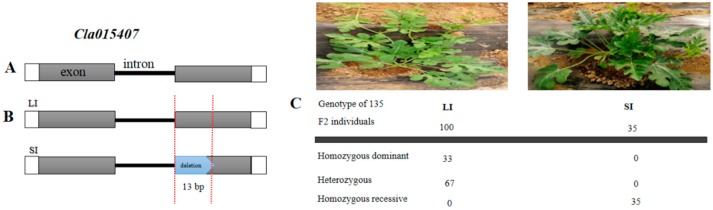
Validation of the candidate gene using an InDel marker. (**A**) The gene structure of *Cla015407*. Gray boxes represent exons and open boxes represent untranslated regions (UTRs), while lines represent introns. (**B**) Confirmation of the deletion in short internode through sequencing. The vertical red dotted line indicates the 13 bp deletion from 502–514 bp. (**C**) Co-segregation of the short internode phenotype and the 13 bp deletion of *Cla015407* in F_2_ population containing 135 individuals. Genotyping by PCR of the 135 individuals revealed that 33 were homozygous dominant (long internode) and 67 individuals were heterozygous, while 35 individuals were homozygous recessive (short internode).

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
