# Peer review of "Molecular Mapping and Candidate Gene Analysis for GA3 Responsive Short Internode in Watermelon (Citrullus lanatus)"

_ijms, 2019, doi:10.3390/ijms21010290_

Round 1

Reviewer 1 Report

This manuscript identified a candidate gene responsible for short internode phenotype of watermelon using BSA-Seq. The result may be important for helping future increase of watermelon productivity. A series of analyses were well performed and the description is almost suitable for publication. However, I have some concerns to be addressed as below.

Line 17. “GA” suddenly appears. Please explain this acronym there as “GA (gibberellin acid)”.

Line 166. “...biosynthetic pathway that activated by...” Please delete “that”.

Line 167. Please cite one or more references after “...in a cal-1 and ap1-1 background”.

Line 174 and Materials and Methods section. There is no explanation and references on “UniProt” and “SMART”. Authors should add them.

Line 176. Candidate gene, Cla015407, contains 208bp deletion. Does this deletion lose a hydroxylase domain? Illustration of domain structure of this gene may aid in understanding this mutation.

Line 237. There is no description on analysis using InDel marker in Materials and Methods section. Authors should add it.

Line 258. Please rephrase “have” with “has”.

Line 276. Please show “Li et al. 2011” by numbering.

Throughout the text, “many” reference numbers may be unsuitable. Please carefully check and correct references. For example, No. 55 cited in line 275 is a paper regarding wheat but not watermelon.

Author Response

The authors thank both the reviewers and the editor. Reviewers had analyzed our manuscript very deeply and had provided important and crucial points that helped in making this paper improved and statistically correct. Such in-depth review would have taken a reasonable amount of time and thus we are highly grateful to the reviewers for helping us improve our paper.

Reviewer 2 Report

The manuscript entitled "Molecular Mapping and Candidate Gene Analysis for GA3 Responsive Short Internode in Watermelon (Citrullus lanatus)" is based on an original research work and there is no doubt that this work is in scope of International Journal of Molecular Sciences.

The introduction is properly composed. The materials and methods section contains the basic requested elements and provide information about the data collection and there analyses. The obtained data are discussed sufficiently.

However, the authors made shortcomings that should be corrected before the publication of this work:

Keywords: there should not be these same words as in title (e.g. watermelon, short internode).

Introduction: Delete lines 89-91, the advantages of this study are presented in conclusions (lines 316-317) Results: In most of the figures there is no need to have 2 numbers after decimal on y-axis (eg. Fig2A number of internodes should be 0, 5, 10 … not 0.00, 5.00, 10.00 … Discussion: Absent of discussion for the cytological analysis results with other studies.

Author Response

[IJMS] Manuscript ID: ijms-677614

General Comments from Authors:

The authors thank both the reviewers and the editor. Reviewers had analyzed our manuscript very deeply and had provided important and crucial points that helped in making this paper improved and statistically correct. Such in-depth review would have taken a reasonable amount of time and thus we are highly grateful to the reviewers for helping us improve our paper.

Response to Reviewer 2 Comments

Point 1: Keywords: there should not be these same words as in title (e.g. watermelon, short internode).

Response 1: The words `watermelon and short internode` has been deleted and candidate gene and cytological analysis has been included.

Point 2: Introduction: Delete lines 89-91, the advantages of this study are presented in conclusions (lines 316-317)

Rseponse 2: The sentence `Our findings will be useful for development of breeding approaches such as gene function analysis and marker assisted selection (MAS) for short internode phenotypes in watermelon`(Line 91-93) has been deleted and we made some correction on the sentence before it.

Point 3: Results: In most of the figures, there is no need to have 2 numbers after decimal on y-axis (eg. Fig2A number of internodes should be 0, 5, 10 … not 0.00, 5.00, 10.00….

Rseponse 3: In in Figure 2 (A, B, D and E) and Figure 6 (A and B), we have deleted the values after decimal (zero, 0.00).

Point 4: Discussion: Absent of discussion for the cytological analysis results with other studies.

Rseponse 4: The discussion on cytological analysis has been supported with the following references.

GA3 is important in determinging plant height by regulating internode cell elongation. In rice, a positive regulator of both GA biosynthesis and GA signaling AtERF11 gene is associated with internode cell elongation and promotes by increasing bioactive GA3 accumulation (Wang et al., 2017). Exogenous GA3 application enhanced internode cell elongation in pea that the dwarf cultivar responds positively to exogenous GA3 (Jones and Roddic, 1977).  Furthermore, other studies have shown that GA3 promotes cell elongation in internode tissue and act as regulators of stem elongation (Ueguchi-Tanaka et al., 2005; Wang et al., 2018). GA3 induced cell elongation should have great significance to rice plants, which in turn develop a low-sensitivity pathway in response to high hormone levels (Tong et al., 2014).

References

Jones T.K.a., Roddic J.G. (1977) Effect of steroidal responses and gibberellic acid on stem elongation on tall and dwarf cultivars of Pisum sativum. New Phytol 79:493-499.

Tong H., Xiao Y., Liu D., Gao S., Liu L., Yin Y., Jin Y., Qian Q., Chu C. (2014) Brassinosteroid regulates cell elongation by modulating gibberellin metabolism in rice. Plant Cell 26:4376-93. DOI: 10.1105/tpc.114.132092.

Ueguchi-Tanaka M., Ashikari M., Nakajima M., Itoh H., Katoh E., Kobayashi M., Chow T.Y., Hsing Y.I., Kitano H., Yamaguchi I., Matsuoka M. (2005) GIBBERELLIN INSENSITIVE DWARF1 encodes a soluble receptor for gibberellin. Nature 437:693-8. DOI: 10.1038/nature04028.

Wang T., Liu L., Wang X., Liang L., Yue J., Li L. (2018) Comparative Analyses of Anatomical Structure, Phytohormone Levels, and Gene Expression Profiles Reveal Potential Dwarfing Mechanisms in Shengyin Bamboo (Phyllostachys edulis f. tubaeformis). Int J Mol Sci 19. DOI: 10.3390/ijms19061697.

Wang Y., Zhao J., Lu W., Deng D. (2017) Gibberellin in plant height control: old player, new story. Plant Cell Rep 36:391-398. DOI: 10.1007/s00299-017-2104-5.